# Fatigue induces long-lasting detrimental changes in motor-skill learning

**Meret Branscheidt[1,2]\*, Panagiotis Kassavetis[1,3,4], Manuel Anaya[1], Davis Rogers[1,5], Han Debra Huang[1], Martin A Lindquist[6], Pablo Celnik[1]\***

[1]The Human Brain Physiology and Stimulation Laboratory, Department of Physical Medicine and Rehabilitation, Johns Hopkins University, Baltimore, Maryland; [2]Clinical Neuroscience Center, University Hospital Zurich, Zurich, Switzerland; [3]Sobell Department of Motor Neuroscience and Movement Disorders, UCL Queen Square Institute of Neurology, University College London, London, United Kingdom; [4]Neurology Department, Boston University, Boston, Massachusetts; [5]The Johns Hopkins University School of Medicine, Baltimore, Maryland; [6]Department of Biostatistics, Johns Hopkins University, Baltimore, Maryland

**Abstract** Fatigue due to physical exertion is a ubiquitous phenomenon in everyday life and especially common in a range of neurological diseases. While the effect of fatigue on limiting skill execution are well known, its influence on learning new skills is unclear. This is of particular interest as it is common practice to train athletes, musicians or perform rehabilitation exercises up to and beyond a point of fatigue. In a series of experiments, we describe how muscle fatigue, defined as degradation of maximum force after exertion, impairs motor-skill learning beyond its effects on task execution. The negative effects on learning are evidenced by impaired task acquisition on subsequent practice days even in the absence of fatigue. Further, we found that this effect is in part mediated centrally and can be alleviated by altering motor cortex function. Thus, the common practice of training while, or beyond, fatigue levels should be carefully reconsidered, since this affects overall long-term skill learning.
DOI: https://doi.org/10.7554/eLife.40578.001

**\*For correspondence:**
mbransc1@jhmi.edu (MB);
pcelnik@jhmi.edu (PC)

**Competing interests:** The authors declare that no competing interests exist.

## Introduction

We know from everyday life that, in order to gain and maintain proficiency, the most critical requirement in a motor skill is practice. Intensive, repetitive training is an essential routine for musicians, artists, surgeons, and athletes. Repetitive practice is also part of rehabilitation approaches to recover function of the motor system and other domains. While repetition improves performance over time, there comes the point when it also causes fatigue and eventual degradation of task execution (*Boyas and Guével, 2011*; *Gandevia et al., 1995b*).

Studies investigating fatigue have made a distinction between fatigue as a cognitive phenomenon and fatigue as a neuromuscular phenomenon (*Janet, 2012*), although this separation can be blurred at times (*Kuppuswamy, 2017*). In neurological conditions, for instance, fatigue has been described as an overall state linked to changes in motor cortex excitability (*Kuppuswamy et al., 2015*). Here, we use the term fatigue to describe the degradation of maximal force output induced through voluntary physical exertion of task-relevant muscles.

Surprisingly little is known about the effects of muscle fatigue on the acquisition of motor skills. The existing literature regarding motor learning under fatigue is mostly limited to a few studies from the 1970–90 s with contradictory results (for a comprehensive overview see *Janet, 2012*). While some studies have reported that participants are unable to learn a motor task under fatigue

**eLife digest** Mastering a new movement requires practice. Intensive and repetitive training is essential for musicians, athletes, or surgeons. It is also important for people undergoing rehabilitation to help them regain normal movements after an illness or injury. Although practice is said to make perfect, there comes the point when it also causes physical fatigue. Fatigue can impair how well a person performs a movement, but its effects on learning a task are less clear.

Now, Branscheidt et al. show that being physically fatigued interferes with learning a new movement skill. In the experiments, volunteers were divided in two groups: the first group had to learn a new motor skill after their hand muscles were physically fatigued, the second group learned the same task without being worn out. The fatigued volunteers had a harder time learning a new motor task both on the day of the task and on the following days, even after they had recovered from the fatigue. The same experiment was repeated, but instead of learning a motor task, the volunteers were asked to learn a sequence of keystrokes. The volunteers in both groups learned this new thinking task easily. This suggests that learning new thinking tasks is not affected by physical fatigue.

Branscheidt et al. also disrupted memory formation in part of the brain that controls movement after volunteers finished learning the motor task using a technique called repetitive transcranial magnetic stimulation. This eliminated the motor learning deficit in the fatigued group. This may suggest that memories formed after fatigue may impair later motor learning and that physical training or rehabilitation that pushes people to work past fatigue may be counterproductive. Further study of these processes may help to develop better training regimens and rehabilitation methods.
DOI: https://doi.org/10.7554/eLife.40578.002

conditions (*Carron and Ferchuk, 1971*, *Thomas et al., 1975*), others have not found fatigue to be detrimental to motor learning (*Cotten et al., 1972 Alderman, 1965*, *Spano and Burke, 1976*).

One often overlooked key challenge in studying motor learning under fatigue is the so-termed 'performance-learning' distinction (*Cahill et al., 2001*; *Kantak and Winstein, 2012*): Performance is usually defined as a temporary effect; for example, how skillful a movement is executed during one training session. In contrast, learning can only be inferred indirectly from performance, by measuring differences in performance over time or tasks (*Kantak and Winstein, 2012*). This distinction is essential because experimental conditions that affect performance do not necessarily have to affect learning. For example, while the performance of rats in the absence of a motivational cue seemed to show no learning in a maze task despite repeated practice, providing a food reward uncovered that they indeed had learned the right path nonetheless (*Tolman and Honzik, 1930*). Thus, it is necessary to separate decreased task performance under fatigue with true effects of fatigue on motor learning. Here, we address this issue by disentangling the effect of muscle fatigue on learning a motor skill from the performance confounder.

In experiment 1 ($N$ = 38), we asked healthy individuals to learn a sequential pinch force task over two days and showed that, even though participants were only fatigued at Day 1, skill learning was impaired on both days. Interestingly, a subgroup of fatigued subjects ($N$ = 12) took two additional days of training with no fatigue to catch up to the skill performance level of the non-fatigued group. In experiment 2 ($N$ = 20), we tested performance on the untrained, unfatigued hand and demonstrated that participants had impaired skill learning in both the fatigued and unfatigued effector. In experiment 3 ($N$ = 45), we replicated the findings of experiment 1 and tested whether the negative effects of fatigue on learning are centrally mediated. We found that disruptive rTMS to the motor cortex (*Cantarero et al., 2013a*; *Huang et al., 2010*) partly alleviates the adverse effects of fatigue on skill learning, suggesting a possible role for maladaptive memory formation under fatigued conditions. Finally, in Experiment 4 ($N$ = 18), we investigated if the observed fatigue effects are domain-specific or also present in another task that is cognitive demanding but requires minimal force control. We found that muscle fatigue did not affect the learning of a ten-sequence element task on Day 1 or 2.

Altogether, our results provide the first evidence that motor fatigue has a domain-specific lasting adverse effect on skill learning. The findings are significant to professions that rely on intensive

physical training to achieve optimal performance. Understanding the effects of fatigue on learning helps the formulation of training and rehabilitation regimens geared to improve motor function.

## Results

### Muscle fatigue has lasting effects on acquisition of force-control demanding motor skill

In the first experiment, we assessed how muscle fatigue influenced skill learning over multiple days. 38 participants trained in a force-control demanding, isometric pinch task for two days; see *Figure 1*. While all subjects were instructed to perform an isometric pinch contraction prior to four bouts of training on Day 1, a subset of participants (Fatigue group (FTG), $N = 20$) performed the contractions until experiencing muscle fatigue (~60% decrement of maximal voluntary contraction. MVC was measured in Newton and monitored by surface electromyographic (EMG) signal). A control group (NoFTG, $N = 18$) contracted the same muscle group at ~5% of MVC over a matched period of time without experiencing force decline. On Day 2 both groups performed the skill task without the induction of fatigue. Skill learning was indexed by a measure that quantifies shifts in the relationship between movement time and accuracy rate (*Reis et al., 2009*). As the relationship between learning rate and skill measure appeared linear, a regression line was fit separately for each day and group. Here the slope of the regression line represents the learning rate (see Materials and methods).

Fatigue was reliably induced during Day 1 in the fatigue group, as shown by decrements of MVC over time. Importantly, MVC always stayed above the force level required to execute the task (up to 40% of MVC; see Appendix 1). All participants improved their ability to execute the task on both days; see *Figure 1*. However, on both days, learning rates for the FTG were significantly reduced when compared to the NoFTG (Day 1: mean slopes NoFTG 0.169 versus FTG 0.038; p=0.01. Day 2: mean slopes NoFTG 0.339 versus FTG 0.083; p=0.03; *Figure 1*). It is important to note that the lower performance of the fatigued group on Day 1 does not allow to make a direct inference about lower motor learning (e.g., because of task differences). Nonetheless, on Day 2 the FTG did not even reach the same execution level the NoFTG subjects had at the end of Day 1 (p=0.01), despite having twice the amount of practice and not been fatigued (i.e. they had similar MVC and were performing within the same force range). Of note, there were no changes in overall learning rate across days for either controls or the fatigued group (Day one versus Day 2 NoFTG: p=0.21, FTG: p=0.77), indicating that learning rates remained low in the FTG.

A separate analysis of the changes in movement time and percentage of correct trials showed that the effect of fatigue on skill performance was due to more errors in the FTG and not to differences in movement time. Counter-intuitively, this higher rate of errors was due to increased force production in the FTG resulting in target overshoot. This was most prominent for the two lower force targets while performance in the highest force target was similar between groups (see Appendix 1).

To assess, how much practice the FTG took to reach similar performance levels as the NoFTG, a subgroup of participants continued training up to four days (NoFTG$_{4D}$ $N = 12$, FTG$_{4D}$ $N = 12$). While Day 1 and two showed the same result with lower learning rates for the FTG$_{4D}$ (Day 1: p<0.01; Day 2: p<0.01), this group reached similar performance levels to the control group only towards the end of Day 3 and on Day 4 (Day 3: p=0.07; Day 4: p=0.09; see *Figure 2*).

Since learning in the FTG was impaired even in the non-fatigued state on Day 2, these results indicate that learning under fatigue conditions has a long-lasting detrimental effect on skill acquisition.

### Fatigue affects performance even in the untrained, non-fatigued effector

Because execution under fatigue is impaired, it is conceivable that this performance confounder masked skill learning. Assessing the transfer of learning to the uninstructed, unfatigued hand provides a unique way to circumvent this challenge. Generalization of motor skills across hands has previously been well characterized, where skill training with one hand results in improved performance in the untrained hand (*Camus et al., 2009*; *Perez et al., 2007*). Thus, in experiment 2, we measured skill execution in the left hand of a new group of 20 participants before and after training with their right fatigued (FTG$_{TRANSFER}$, $N = 10$) or non-fatigued hand (NoFTG$_{TRANSFER}$, $N = 10$).

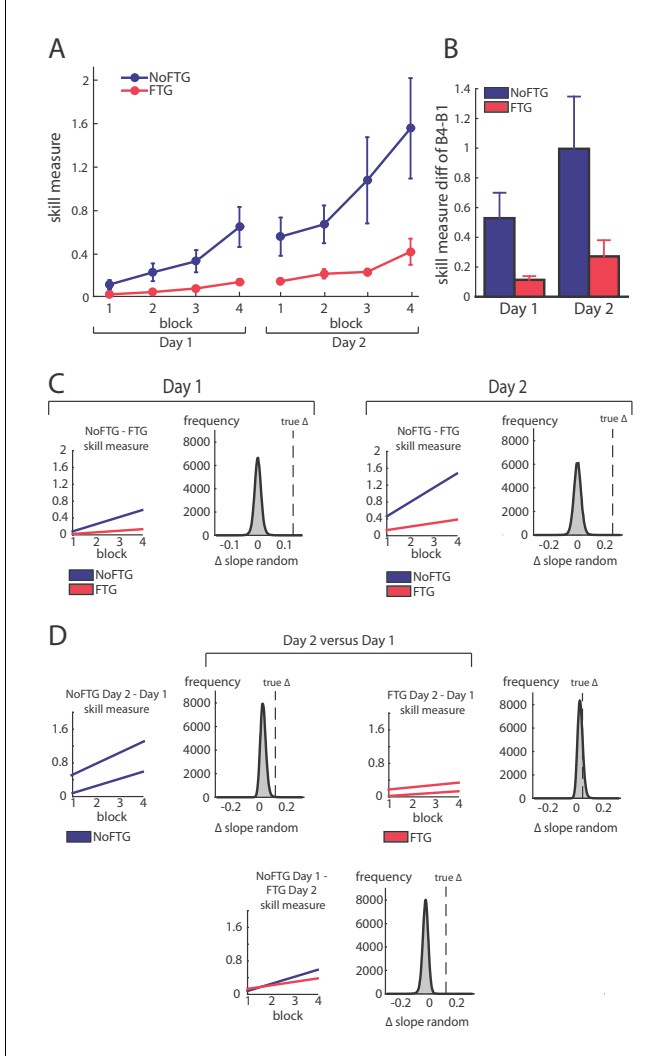

**Figure 1.** Comparison of skill acquisition in an isometric pinch task between fatigued and non-fatigued participants. Panel (**A**) shows changes in skill measure over the course of four training blocks on two consecutive days for both groups (NoFTG = blue; FTG = red). While both groups improved task execution, the FTG had a lower performance level on Day 1 and Day 2 when they were not fatigued compared to controls. Note that the skill performance in block 4 of Day 2 in the FTG remained below the level of NoFTG at the end of Day 1. Panel (**B**) shows the difference in performance between block 4 to block one for both groups on each day. Panel (**C**) shows the differences in learning rates for Day 1 and Day 2 between groups. We compared the learning rate of both groups by first fitting a robust linear regression model to the individual data of each group (line plots). To test if there was a true difference in learning rates, we calculated the Δ of the average slope of the fitted model (regression coefficients). We then used permutation testing to generate a distribution of Δ slopes of randomly generated groups (grey histoplots). The Null hypothesis was rejected when less than 5% of the generated Δ slopes exceed the true Δ slope (dashed vertical lines, see also Materials and methods). In Panel (**D**) learning rates are compared within groups across days (Day 2 versus Day 1). There were no significant differences in learning rates across days in both groups.

DOI: https://doi.org/10.7554/eLife.40578.003

Similar to experiment 1, right hand learning rates over the four blocks were lower in participants that performed the task under fatigue compared to controls (mean slope NoFTG$_{TRANSFER}$ 0.03 versus FTG$_{TRANSFER}$ 0.008; p=0.01; *Figure 3*). As expected, prior to training, the skill measure of the left hand was similar between groups ($t_{18}$ = −0.157, p=0.88). After training with the right hand, performance of the left hand was significantly lower in the FTG$_{TRANSFER}$ compared to the NoFTG$_{TRANSFER}$

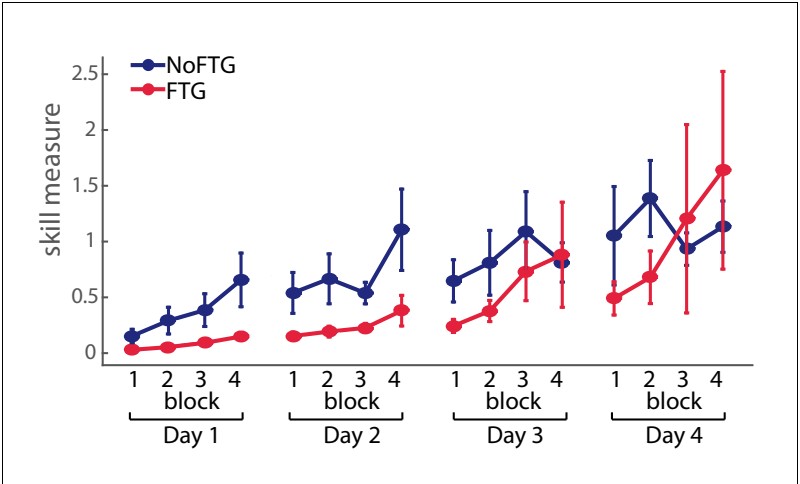

**Figure 2.** Comparison of skill execution between FTG and NoFTG over the course of four days. Note that the FTG (red) showed lower skill levels at Day 1 and 2 compared to the NoFTG (blue) and only reached similar levels to controls at the end of Day 3 and on Day 4.
DOI: https://doi.org/10.7554/eLife.40578.004

(mean $\Delta$block2-block1 NoFTG$_{TRANSFER}$ 0.133 $\pm$ 0.036 versus FTG$_{TRANSFER}$ 0.018 $\pm$ 0.003; p<0.01; *Figure 3*).

Based on the work of Tolman and Honzik (*Tolman and Honzik, 1930*) differences in skill performance between the fatigued and the control groups could arise from impeded execution in the right hand, which could theoretically mask similar underlying learning rates in both groups. If this would be the case, then the unfatigued effector in both groups should show comparable skill levels. However, our results indicate that fatigue did indeed affect learning and that the performance confounder did not just mask the expression of learning.

Importantly, the poor performance in the untrained hand suggests that fatigue impairs central motor-skill learning mechanisms beyond any potential adverse effect within the fatigued effector.

## Long-lasting detrimental effects of fatigue on learning are centrally mediated

To determine whether the effect of fatigue on learning is centrally mediated, we interfered with primary motor cortex processes thought to be involved in skill retention in a new group of participants (*Galea et al., 2011*; *Muellbacher et al., 2002*; *Reis et al., 2009*; *Richardson et al., 2006*). To this end, in experiment 3 we used disruptive rTMS (repetitive transcranial magnetic stimulation) over the primary motor cortex (M1) after task training on Day 1 (FTG$_{M1}$, N = 15) (*Cantarero et al., 2013a*; *Huang et al., 2010*). To control for potential non-specific effects of rTMS, we also tested a fatigued and a non-fatigued group with TMS applied over the parietal interhemispheric fissure (Pz, according to 10–20 system; FTG$_{SHAM}$, N = 15 and NoFTG$_{SHAM}$, N = 10).

The permutation test showed that learning rates of both fatigued groups were smaller compared to controls, but similar to each other on Day 1 (mean slope NoFTG$_{SHAM}$ 0.049, FTG$_{SHAM}$ 0.02, FTG$_{M1}$ 0.016; NoFTG$_{SHAM}$ versus FTG$_{SHAM}$ p=0.01, NoFTG$_{SHAM}$ versus FTG$_{M1}$ p<0.01, FTG$_{SHAM}$ versus FTG$_{M1}$ p=0.73; *Figure 4*). On Day 2, consistent with experiment 1 and 2, the learning rate was still smaller in FTG$_{SHAM}$ compared to the NoFTG$_{SHAM}$ control. However, the learning rate of the FTG$_{M1}$ group was *not* significantly different from the NoFTG$_{SHAM}$, but significantly different from FTG$_{SHAM}$ (mean slope NoFTG$_{SHAM}$ 0.04, FTG$_{SHAM}$ 0.022, FTG$_{M1}$ 0.042; NoFTG$_{SHAM}$ versus FTG$_{SHAM}$ p=0.04, NoFTG$_{SHAM}$ versus FTG$_{M1}$ p=0.47, FTG$_{SHAM}$ versus FTG$_{M1}$ p=0.03). Of note, comparing learning rates across days within groups, we found a significant difference for FTG$_{M1}$ (Day one versus Day 2, FTG$_{M1}$ p=0.04), but no difference for the other two groups (Day 1 versus Day 2, NoFTG$_{SHAM}$ p=0.94, FTG$_{SHAM}$ p=0.88).

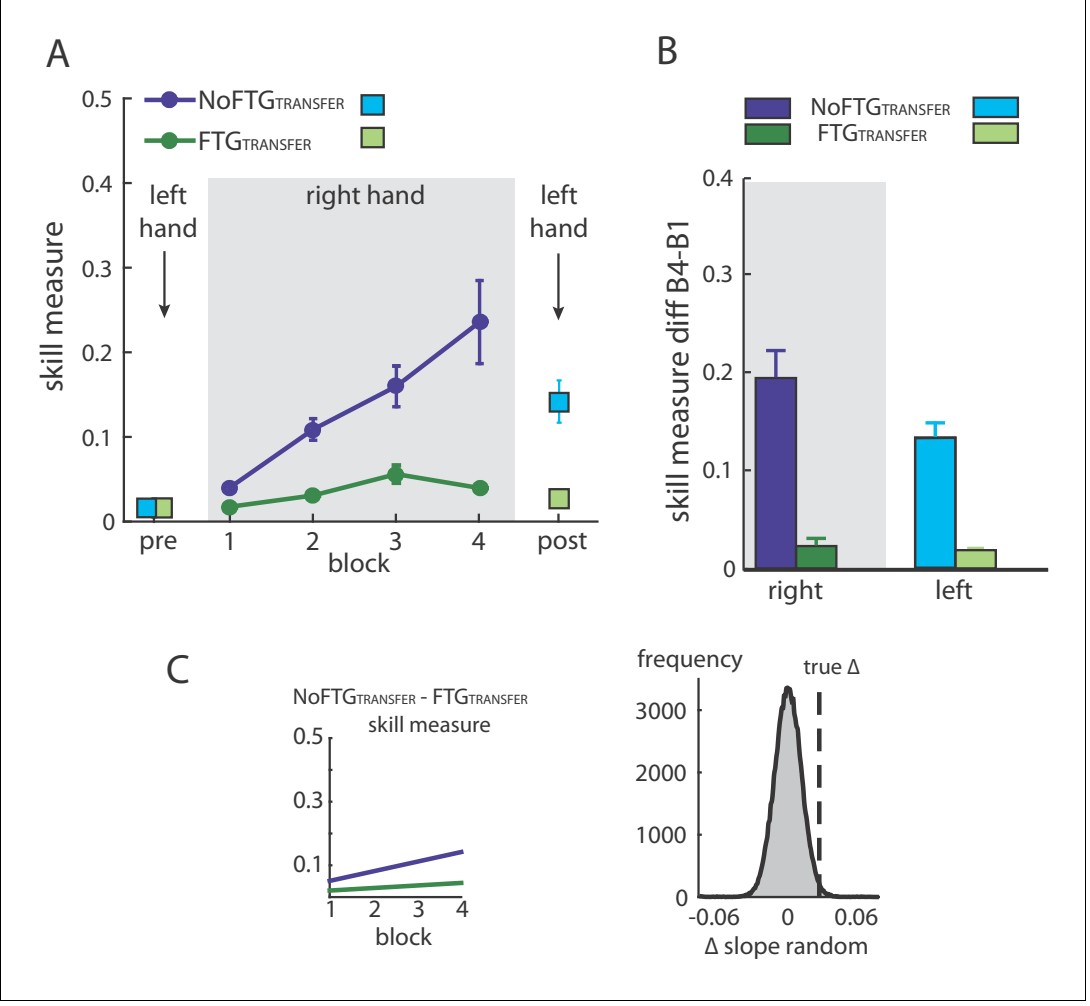

**Figure 3.** Intermanual transfer of learning in fatigued versus non-fatigued participants. Panel (**A**) shows changes in the skill measure over the course of four blocks during one day of training (NoFTG$_{TRANSFER}$ = dark blue, FTG$_{TRANSFER}$ = dark green). Before and after the training, both groups performed 15 trials of the pinch force task with their left hand (FTG$_{TRANSFER}$ = light blue square, NoFTG$_{TRANSFER}$ = light green square). Note that, while both groups improved skill performance over time, the FTG$_{TRANSFER}$ group had a lower performance level, consistent with experiment 1, in both the fatigued and non-fatigued effector. Panel (**B**) shows the difference in performance between block 4 to block one for the right hand and block 2 - block one for the left hand in both groups. Performance in the left hand was significantly lower in the FTG$_{TRANSFER}$ compared to NoFTG$_{TRANSFER}$. Panel (**C**) shows the learning rates and the true Δ slope in comparison to randomly generated Δ slopes after permutation. Similar to experiment 1, controls showed higher learning rates than the fatigued group.
DOI: https://doi.org/10.7554/eLife.40578.005

Together, these results show that disruption of M1 function after training diminished the detrimental effects of fatigue on motor-skill learning. This indicates that the long-lasting effects of fatigue on learning are at least partly centrally mediated and linked to motor memory formation.

## Fatigue does not impair a sequence learning task

To determine whether the observed muscle fatigue effects are specific to tasks with high motor control demand (e.g., accurate force control to complete the sequence) vs. a task with more cognitive demands, we performed an additional control. Here, 18 healthy participants performed a 10-element, finger sequence task by simply pressing the correct key on a computer keyboard after being fatigued (FTG$_{SEQUENCE}$, N = 9) or not (NoFTG$_{SEQUENCE}$, N = 9) as done in the prior experiments (for visualization of the study design the Materials and methods and Appendix 1). We found that fatigue

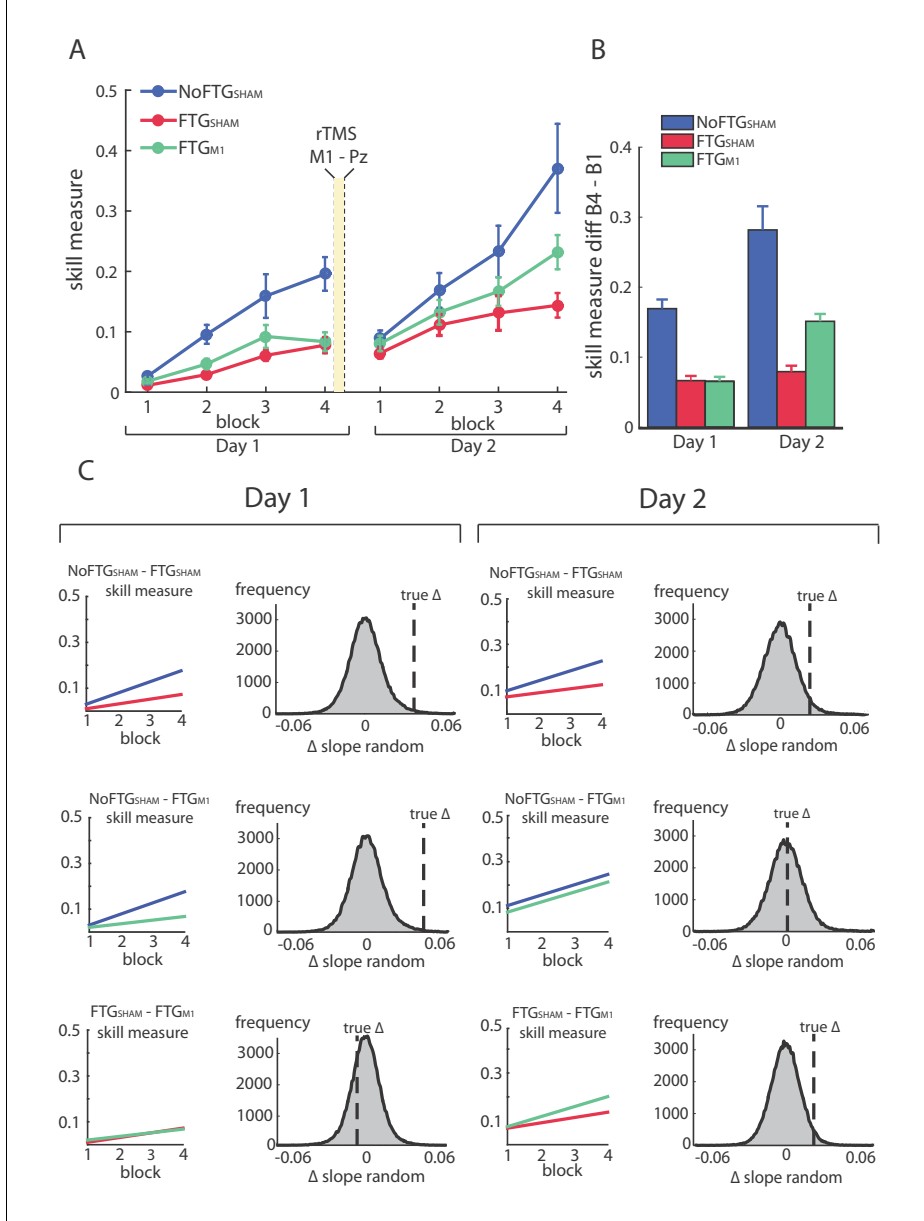

**Figure 4.** Disrupting M1 at the end of Day one training reduces the impaired skill acquisition on Day 2. Panel (**A**) shows changes in the skill measure over four training blocks on two consecutive days for all groups (NoFTG$_{SHAM}$ = blue, FTG$_{SHAM}$ = red, FTG$_{M1}$ = green). Note that the FTG$_{M1}$ experienced higher improvement of performance in Day 2 compared to the FTG$_{SHAM}$. Panel (**B**) shows the difference in performance between block 4 to block one for all groups on each day. Panel (**C**) shows the comparison of learning rates for all groups on Day 1 and 2. On both days, the unfatigued control group showed a higher learning rate than the FTG$_{SHAM}$ group (this group received sham stimulation at the end of Day 1). Second row: In contrast, there was no significant difference in learning rates between the FTG$_{M1}$ group and the control group on Day 2 (in this group, M1 function was disrupted at the end of Day 1 using rTMS), while learning rates on Day 1 were lower. Third row: Both groups showed similar performance on Day 1, but higher learning rates were evidenced on Day 2 in the FTG$_{M1}$ group compared to the FTG$_{SHAM}$ group.

DOI: https://doi.org/10.7554/eLife.40578.006

prior to the sequence learning did not result in different performance compared to the non-fatigued control group on Day 1 or Day 2. Both groups had less errors in block 4 than block one in both days, with no group difference on either day (Day 1: block: $F_{(3,16)} = 4.474$, p=0.021; group: $F_{(1,16)} = 0.329$, p=0.574), block*group: $F_{(3,16)} = 1.458$, p=0.79; Day 2: block: $F_{(3,16)} = 5.363$, p=0.034; group: $F_{(1,16)} = 0.535$, p=0.475), block*group: $F_{(3,16)} = 2.603$, p=0.126). The same was true for movement times. Participants decreased their times from block one to block four, but there was no group differences (Day 1: block: $F_{(3,16)} = 146.34$, p<0.001; group: $F_{(1,16)} = 0.498$, p=0.49), block*-group: $F_{(3,16)} = 0.169$, p=0.972; Day 2: block: $F_{(3,16)} = 11.31$, p=0.004; group: $F_{(1,16)} = 1.106$, p=0.309), block*group: $F_{(3,16)} = 4.192$, p=0.057; *Figure 5*).

These results indicate that the detrimental effects of muscle fatigue on learning are specific to skill tasks that required fine force-control, but not in more cognitive-demanding tasks.

## Fatigue in the absence of training does not impair learning on a subsequent day

To ensure that the results from the previous experiments were not due to prolonged physical manifestations of fatigue on Day 2, we fatigued a new group of participants on Day 1 but did not expose them to the pinch force task until the second day ($FTG_{SKILL-DAY2}$, $N$ = 5). On Day 2, this group showed similar learning rates when compared to the control group on Day 1 (mean slope $FTG_{SKILL-DAY2}$ 0.049; p=0.50; see Appendix 1).

## Discussion

We investigated the effects of fatigue on motor-skill learning while excluding performance confounders. As expected, muscle fatigue resulted in lower levels of performance immediately after. However, training under conditions of fatigue on Day 1 also affected learning in subsequent training sessions, even in the absence of fatigue. This effect was of such magnitude that at the end of the 2$^{nd}$ practice session the FTG group achieved only 68% of the performance level of the NoFTG group at the end of the first session. This detrimental effect persisted for almost two complete additional training sessions before performance arrived at equal levels at the end of Day 3. The deleterious effect of fatigue on learning was also found on the opposite, not fatigued hand and reverted by a M1 rTMS protocol known to interfere with memory retention (*Cantarero et al., 2013a*). Further, the effects of muscle fatigue on skill learning were not present when fatigue was induced but not followed by training and when the training involved a sequence task with significant cognitive load but not force-control demand. Our results provide evidence for a centrally-mediated, domain-specific deleterious effect of fatigue on motor-skill learning beyond impairment in execution.

The observed differences in motor task performance were mostly driven by lower accuracy rates in the fatigue groups, while movement times were similar across groups. Interestingly, the reduced accuracy was due to a larger number of overshooting errors, which required higher force despite the presence of muscle fatigue. This seemingly counterintuitive finding may be due to sensory attenuation. Sensory attenuation can be defined as the precision with which sensory input from self-generated movement is perceived (*Shergill et al., 2003*). Under normal circumstances, there is a degree of sensory attenuation associated with voluntary muscle contraction, with lesser attenuation seen for higher force levels (*Walsh et al., 2011*). However, sustained isometric contraction similar to our fatiguing task has been described to directly affect the activity of primary muscle spindle afferents as a consequence of thixotropic properties of intrafusal muscle fibres (*Luu et al., 2011*). As a result, an underestimation of the absolute applied forces (especially in the lower force range) based on the attenuation of sensory consequences could be expected (*Luu et al., 2011*, *Brooks et al., 2013*). A common experience of this effect in everyday life is the Kohnstamm's phenomenon, where preconditioning of the muscle spindles with isometric contraction has a significant effect on position sense and sense of effort (*Hagbarth and Nordin, 1998*). Therefore, it is conceivable that changes in sensory attenuation induced by fatigue led to impaired force control on Day 1. In support of this, Park et al found in a finger force-matching task that after fatigue force production was higher compared to the instructed reference force at low force levels (15% of MVC) but not for medium force levels (35% MVC) (*Park et al., 2007*). In spite of this, it should be noted that other measures of force production show opposite results with increased perceived force under sustained fatigue conditions (*Pageaux and Lepers, 2016*).

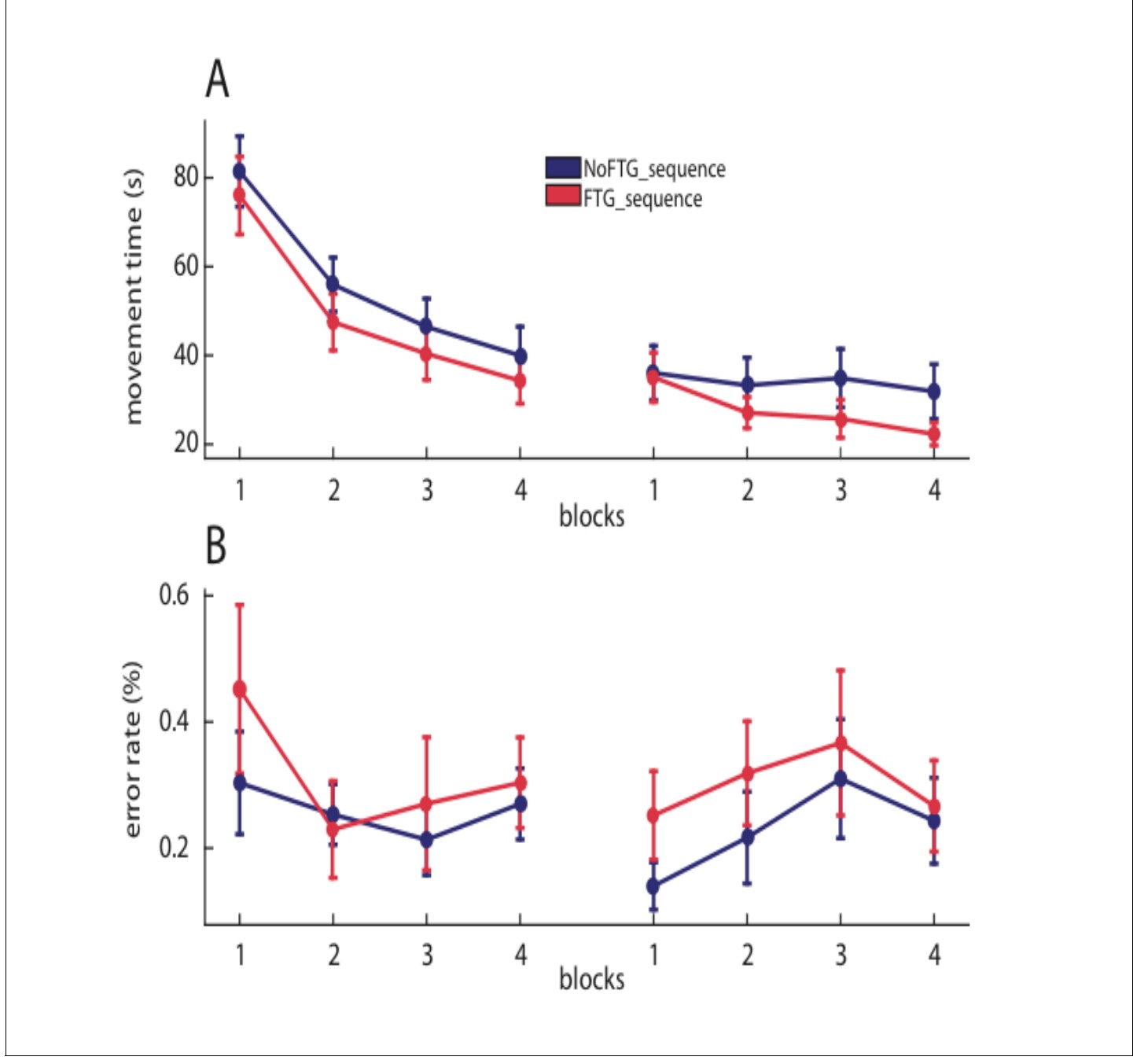

**Figure 5.** Muscle fatigue does not impair sequence learning. Panel (**A**) shows changes in movement time over the course of four blocks on two consecutive days for the non-fatigued versus the fatigued group (NoFTG$_{SEQUENCE}$ and FTG$_{SEQUENCE}$). Both groups performed similar on the task on both days. Panel (**B**) shows the difference in error rate for both groups on both days. Again, we found no differences in performance between the groups.

DOI: https://doi.org/10.7554/eLife.40578.007

Alternatively, it has recently been suggested that fatigue might distort visual perception. This has been shown from both egocentric perspective (*Witt et al., 2004*) and allocentric perspective (*Kuppuswamy et al., 2016*). For example, participants who experienced high-level physical exertion systematically overestimated the length of a visually presented line, which could result in target over-shooting (*Kuppuswamy et al., 2016*).

How muscle fatigue exerts its effect on subsequent training sessions and takes multiple days to wash out is not completely clear. We propose that subsequent learning is impaired because specific central mechanisms are affected by the original training under fatigue. For instance, it is conceivable that aspects of task performance under fatigue (e.g., changes in the pattern and intensity of muscle activation) are remembered and recalled during subsequent practice even though the system properties have changed between Day 1 and Day 2. Muscular fatigue induces peripheral as well as central changes, such as decreases of motor cortical drive to the muscles followed by depression of motor cortex excitability (*Gandevia et al., 1995a*; *Taylor et al., 2000*; *Zanette et al., 1995*). This is particularly interesting as motor cortex function has been attributed an essential role for skill learning and retention (*Kawai et al., 2015*; *Shmuelof and Krakauer, 2011*; *Reis et al., 2009*). Supporting evidence to central mechanisms also comes from Takahashi and colleagues who found that subjects produced higher forces during a second exposure to a force field if they had practiced the task under fatigue compared to practice in a non-fatigued condition (*Takahashi et al., 2006*). The results of experiment 3 also indicate a central mechanism. Here, we found that participants who were fatigued on Day 1 but received disruptive rTMS over the motor cortex, known to affect retention after training (*Cantarero et al., 2013a*), expressed higher learning rates on the subsequent day than participants who only received rTMS over a cortical control site. Indeed, after rTMS no differences in learning rate could be found when compared to non-fatigued controls on Day 2. Therefore, our results indicate that learning under fatigue can lead to the formation of specific memories that are not helpful to subsequent training in a non-fatigued state, slowing down overall learning. This muscle fatigue effect seems to be specific for motor tasks with significant force-control demand, but not for cognitively challenging tasks where force control is not necessary.

The persistent limited skill acquisition following training under fatigue may be attributed to time-dependent differences in the contribution of explicit and automatized memory-based processes (*Taylor and Ivry, 2012*). Explicit strategic processes generally have been associated with early stages of learning, when the difference between the movement goal and the chosen motor command is large (*Taylor and Ivry, 2012*). In this stage, exploration of the manifold is believed to lead toward selection of the optimal (or close to optimal) solution strategy, which is then followed by a gradual refinement of the chosen action sequence and smaller changes in behavior. Thus, it is conceivable that aspects of task performance under fatigue are indeed retained and perceived as the optimal movement strategy on the second day, leading to slower performance gains. In other words, participants that learned the 'optimal motor commands for the fatigued state' may lack a de novo exploratory stage in the subsequent exposures to the task, resulting in continued use of a strategy that is suboptimal for learning which, in turn, results in lower learning rates.

In our understanding, the observed result cannot be explained by context specificity. It has been argued that, because fatigue leads to changes in the pattern and intensity of muscle activation as well as to changes in sensory feedback, what has been learned under fatigue can only show limited transfer to performance in the unfatigued state and vice versa (*Barnett et al., 1973*; *Janet, 2012*). While limited transfer can explain an initially lower performance of the fatigued group on Day 2, it cannot account for the lower learning rates throughout Day 2 and sustained effect on performance up to Day 3.

## Conclusion

We tested motor learning of a skill task under conditions of fatigue. We found that learning in a fatigued state results in detrimental effects on overall task acquisition. These phenomena are present above and beyond the deleterious consequences of fatigue on performance and appears to be domain-specific, and at least in part, centrally mediated. The deleterious fatigue effect was of such a magnitude that took participants twice as much time to reach the level of performance of individuals who learned the task de novo in non-fatigue conditions. These observations need to be carefully considered when designing training protocols such as in sports or musical performance as well as for rehabilitation programs. While conditions of fatigue during sports or performing arts can occur by chance or by overachieving attitudes, rehabilitation programs are particularly at risk because patients with neurological conditions such as those following stroke or multiple sclerosis frequently experience fatigue.

## Materials and methods

### Participants

A total of 121 healthy participants were recruited from two centers (Johns Hopkins University and University College London). None of the participants suffered from any neurological or psychiatric disorder, nor were they taking any centrally-acting prescribed medication. The experiments were approved by the respective ethics boards at Johns Hopkins School of Medicine Institutional Review Board and the North West London Research Ethics Committee in accordance with the Declaration of Helsinki and written informed consent as well as consent to publish was obtained from all participants (ethics board number 00077792). For the first experiment, the sample size was chosen in line with previously reported effect sizes in motor-skill learning studies (*Cantarero et al., 2013a*; *Cantarero et al., 2013b*; *Reis et al., 2009*).

### Motor task

For each experiment, participants were seated in front of a computer monitor and given a force transducer to hold between the thumb and index fingers of their dominant hand. During each trial, participants were instructed to produce isometric pinch presses at different force levels to control the motion of a cursor displayed on the screen. Increasing force resulted in the cursor moving horizontally to the right. Participants were instructed to increase and decrease their pinching force to navigate the cursor through the following sequence: start-gate1-start-gate2-start-gate3-start-gate4-start-end; see *Figure 6*. The cursor movement followed logarithmic transduction of the applied pinch force as described in previous studies (*Reis et al., 2009*). This task has been widely used to study skill learning (e.g., *Reis et al., 2009*; *Cantarero et al., 2013a*). It involves two components of learning, speed and accuracy, which we could explore independently (see also Data Analysis and Appendix 1). While the chosen task allows for detailed behavioral assessment of changes underlying learning, due to differences in muscle strengths across fingers involved in the task, the behavioral assessment is potentially less suited to analyze independent mechanisms of peripheral and central fatigue within the hand muscles involved.

### Sequence learning task

Participants were seated in front of a computer screen and presented with a horizontal display of five square stimuli ('G', 'H',' J', 'B' or 'N') with one highlighted in green. Subjects were instructed to press as fast as possible the corresponding computer key on a desktop keyboard with their index finger. The next element of the sequence was only presented after the correct key response. If an incorrect key response was pressed, the sequence was paused and only resumed once the appropriate key response was made. Each sequence trial started with the presentation of a 'go' cue. Participants were exposed to the same 10-element sequence on each trial and had to perform 30 trials in total to complete one of four blocks.

### Design

#### Experiment 1: Determining the effect of fatigue on temporal aspects of motor-skill learning

38 participants (23 women, mean age 22.2, ±2 years, all right-handed) were recruited and randomly assigned to one of two groups, a fatigued group (FTG, *N* = 20) and a non-fatigue group (NoFTG, *N* = 18). All participants underwent ~45 min testing sessions on two or four consecutive days; see *Figure 6*. Sessions took place between 9 a.m. and 6 p.m. and were separated by 24 hr (±1 hr). Morning and afternoon sessions were counterbalanced between groups and subjects performed both sessions at similar times. Each day, both groups performed the isometric pinching task (see motor task) for four blocks of 30 trials each. At the start and the end of each experimentation sessions, participants in both groups were asked to press the force transducer with their maximum force for 5 s in order to assess the maximum voluntary contraction (MVC). On Day 1, the FTG was instructed to sustain MVC until the produced force dropped to the level of the upper limit for gate 2, the target that requires the largest force production. Thus, the induction of fatigue always stayed above the force level needed for task execution. Time to fatigue was 68.91, SD 32.23 s on average. To counterbalance the amount of time of the fatigue induction, the non-fatigue group was asked to sustain 5% of

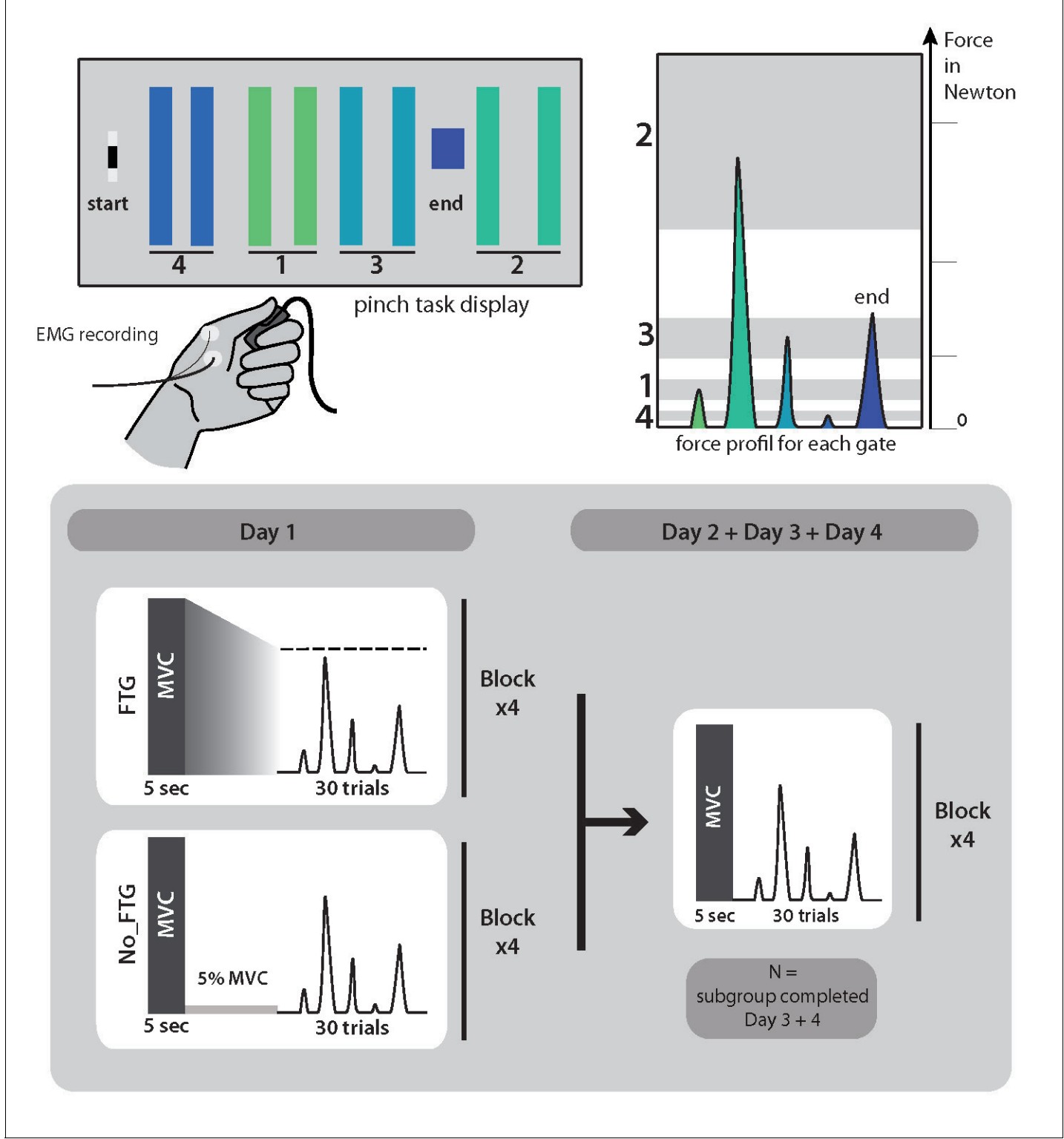

**Figure 6.** Pinch force skill task and study design for experiment 1.
DOI: https://doi.org/10.7554/eLife.40578.008

their MVC over a matched period of time. On Day 2, the design was identical for both groups: assessment of MVC at the start and end of the session, with four blocks of skill task without a break in between. To determine how long fatigue influenced motor-skill learning, a subgroup (N = 24, FTG and NoFTG both N = 12) participated in two extra days of experimentation. The design in Days 3 and 4 was identical to Day 2 for both groups, see *Figure 6*.

### Experiment 2: Determining the effects of fatigue on motor-skill learning measured by transfer of skill

To investigate motor-skill learning within one session while avoiding the execution confounder, we assessed inter-manual skill transfer in the unfatigued, untrained left hand after participants trained with the right hand with or without fatigue. 20 participants were recruited to take part in experiment 2 (17 women, mean age 20.1 ± 3.0 years, all right-handed). Participants were again randomly assigned to either a fatigued or a non-fatigued group (FTG$_{TRANSFER}$, N = 10; NoFTG$_{TRANSFER}$, N = 10) and tested in a single session. The task design was identical to the first day in experiment 1. Additionally, at the beginning and end of the session, participants completed one block of 15 trials with their unfatigued left hand; for visualization of the study design see *Appendix 1—figure 3*.

### Experiment 3: Testing whether the effects of fatigue on learning are centrally mediated

In experiment 3, a new group of 40 healthy participants was recruited (15 women, mean age 23.5 ± 2.9 years, four left-handed, one ambidextrous) and randomly assigned to one of three groups: a non-fatigue group (NoFTG$_{SHAM}$, N = 10), or one of two Fatigue groups (FTG$_{SHAM}$, N = 15 or FTG$_{M1}$, N = 15). The overall study design for all three groups was similar to experiment 1; for visualization of the study design see *Appendix 1—figure 4*. In addition, at the end of Day 1, participants received depotentiation TMS (DePo) either over their registered 'M1 hot-spot' (FTG$_{M1}$; see also TMS section) or a control location (FTG$_{SHAM}$ and NoFTG$_{SHAM}$). The depotentiation stimulation is a shorter form of continuous theta burst stimulation with TMS which has been shown to reverse potentiating plasticity (*Huang et al., 2010*; *Huang et al., 2005*; *Huang et al., 2008*) and disrupt skill retention (*Cantarero et al., 2013a*). DePo stimulation was administered in a double-blind fashion. A researcher not involved in the behavioral portion of the study delivered the stimulation, while those conducting the behavioral training were blinded to the stimulation location and protocol.

### TMS

Transcranial magnetic stimulation (TMS) was administered with a figure-eight coil (wing diameter = 70 mm) connected to a Magstim 200 stimulator (Magstim, UK). Using TMS, We located the 'hot-spot' of the abductor pollicis brevis muscle in the task-relevant hand at rest according to standardized procedures (*Chen et al., 2008*; *Rossini et al., 1994*). The stimulus intensity that elicited a motor evoked potential (MEP) with a peak-to-peak amplitude of approximately 1 mV was established (Stimulus intensity 1 mV, S1mV) to assess corticomotor excitability. Then 18 MEPs were recorded using the same intensity before the task, directly after the task, and after depotentiation on Day 1 as well as before and after the task at Day 2. The parameters for depotentiation were based on previous reports (*Cantarero et al., 2013a*; *Huang et al., 2010*), consisted of bursts of three pulses at 50 Hz repeated at 200 ms intervals at an intensity of 70% of the rMT for 10 s. Using the 10–20 electroencephalogram coordinate system, Pz was used as a control stimulation location (SHAM stimulation). Stereotactic neuronavigation (BrainSight, Rogue Research, Montreal, Quebec, Canada) was used to track coil position within sessions. EMG activity from the abductor pollicis brevis muscle was recorded using surface electrodes taped in a belly-tendon orientation. Data were recorded with an AMT-8 (Bortec Biomedical Ltd; sampling rate 5000 Hz, amplification 1000x, band-pass filtering 10–1000 Hz) and saved for offline analysis.

### Experiment 4: Determining the effects of fatigue on a sequence learning task

Learning the motor-skill task in experiment 1–3 involves the acquisition of knowledge of a logarithmic force-distance sensorimotor map as well as learning to produce the correct sequence of forces to reach the different targets (*Spampinato and Celnik, 2017*). To further understand if muscle

fatigue influences both of these aspects of skill learning, we added a 10-element sequence task that is cognitively challenging, but has minimal force demands. 18 participants (11 women, mean age 27.5 ± 9.1 years, all right-handed) were randomly assigned to a fatigued or a non-fatigued group (FTG$_{SEQUENCE}$, $N = 9$; NoFTG$_{SEQUENCE}$, $N = 9$). The experimental set-up was identical to Experiment 1, but instead of the pinch force skill task, both groups trained on the 10-element sequence task after fatigue or the control isometric contraction; for visualization of the study design see *Appendix 1—figure 5*.

## Data analysis
For analysis of MVC see Appendix 1.

## Analyzing movement time/Error rate
For each trial in the skill task, movement time and error rate were recorded: movement time was defined as the duration from movement onset (forced controlled cursor leaving start position) to reaching the end gate. Error rate was defined as the percentage of trials per block in which participants under- or overshot at least one of the five targets. For the skill learning task, movement time was defined as the duration between the first and the 10$^{th}$ correct key press. Error rate was defined as the percentage of trials per block (number of errorless sequences per block).

## All experiments
Movement time and error rate were compared using rmANOVA with the within-subject factor 'block' (four levels: b1, b2, b3 and b4) and the between-subject factor 'group' (Exp. 1 and 2: two levels, Exp. 3: three levels) for each single day (see Appendix 1).

## Analyzing motor skill
To quantify motor performance, we calculated a skill measure, composed of movement time and error rate. As done in prior studies, the skill measure was calculated as: a=(1- error rate)/ [error rate (ln(movement time)b)], where b is 5.424 as predefined for this particular task in prior studies (*Reis et al., 2009*; *Cantarero et al., 2013b*; *Mawase et al., 2017*; *Spampinato and Celnik, 2017*).

To study learning (rate of change in performance) during the motor task, we plotted the number of blocks on the x-axis and the skill measured on the y-axis. As the relationship was roughly linear, we fit a robust linear regression model (e.g., f(x)=c*x + b) for each group; the robust function disregards outliers by estimating an iteratively reweighted least square algorithm. This provided a more parsimonious model than ANOVA, providing a single easily interpretable measure of learning rate given by the slope c. We were particularly interested in measuring and comparing the learning rates between days and groups. When only two blocks were tested (i.e. the left hand in experiment 2) we took the difference between block2 – block1 as a measure of learning rate.

Differences in learning rates for each experiment were assessed using a permutation testing procedure. Assuming the null hypothesis of no group difference, participants were randomly reassigned to the two groups, and the difference in regression coefficients between the resampled groups was computed. This procedure was repeated 10,000 times, allowing us to generate a null distribution for the difference between regression coefficients assuming no group differences. The proportion of resampled values that exceeded the true observed difference was used to compute p-values and determine statistical significance. Under the null hypothesis, the true difference in learning rates between the two groups should lie within the distribution of these randomly generated differences, with extreme values providing evidence against the null hypothesis.

Prior to application of any parametric tests, the normality of the dependent variables was assessed using Shapiro-Wilk tests and quantile-quantile plots. A log-transformation was applied to correct for any non-normal data. All ANOVA results were Greenhouse-Geisser corrected if the assumption of sphericity was violated. Student's t-test was used to assess group differences. Results were considered significant at p<0.05, and Bonferroni correction was applied to correct for multiple comparisons. All data are expressed as mean ±standard error unless stated otherwise. Statistical analyses were performed using SPSS 22.0 and custom-written MATLAB routines. Data and custom-written code are available at *Branscheidt (2018)*.

## Acknowledgments

We thank Claudia Ammann for conducting the depotentiation stimulation.

## Additional information

### Funding
No external funding was received for this work.

### Author contributions
Meret Branscheidt, Conceptualization, Data curation, Formal analysis, Validation, Investigation, Visualization, Writing—original draft, Writing—review and editing; Panagiotis Kassavetis, Conceptualization, Data curation, Investigation, Methodology, Writing—review and editing; Manuel Anaya, Data curation, Writing—review and editing; Davis Rogers, Data curation, Investigation, Writing—review and editing; Han Debra Huang, Data curation, Validation; Martin A Lindquist, Formal analysis, Writing—review and editing; Pablo Celnik, Conceptualization, Supervision, Methodology, Writing—review and editing

### Author ORCIDs
Meret Branscheidt (iD) http://orcid.org/0000-0002-4008-6916

### Ethics
Human subjects: The experiments were approved by the respective ethics boards at Johns Hopkins School of Medicine Institutional Review Board and the North West London Research Ethics Committee in accordance to the Declaration of Helsinki, and written informed consent was obtained from all participants (ethics board number 00077792).

### Decision letter and Author response
Decision letter https://doi.org/10.7554/eLife.40578.021
Author response https://doi.org/10.7554/eLife.40578.022

## Additional files

### Supplementary files
• Transparent reporting form
DOI: https://doi.org/10.7554/eLife.40578.009

### Data availability
The full data-set of this study is available at (https://osf.io/ypxfg/).

The following dataset was generated:

| Author(s) | Year | Dataset title | Dataset URL | Database and Identifier |
| --- | --- | --- | --- | --- |
| Branscheidt M | 2018 | Motor learning under fatigue | https://osf.io/ypxfg/ | Open Science Framework, ypxfg |

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

# Appendix 1

DOI: https://doi.org/10.7554/eLife.40578.010

## Analysis of MVC

MVC was expressed as the average absolute force applied on the force transducer for 5 s. Additionally, we recorded surface EMG from the first dorsal interosseous and the adductor pollicis brevis.

### Experiment 1

To ensure that the chosen condition was reliably inducing fatigue over the whole duration of the task, we assessed the decrease of MVC on Day one using repeated-measures ANOVA (rmANOVA) with the within-subject factor 'time' (five levels: preB1, preB2, preB3, preB4, postB4) for the fatigued group. On Day 2, we assessed changes in MVC using an rmANOVA with the within-subject factor 'time' (two levels: pre, post) and the between-subject factor group (two levels: NoFTG, FTG).

### Experiment 2

We compared differences in MVC before and after the experiment, denoted $\Delta$MVC(pre-post), between the two groups ($FTG_{TRANSFER}$ versus $NoFTG_{TRANSFER}$) using a one-way ANOVA.

### Experiment 3, 4 and control condition

Differences in $\Delta$ MVC(pre-post) between days and groups were determined with rmANOVA with the within-subject factor 'day' (two levels: Day one and Day 2) and the between-subject factor 'group' (three levels for Experiment 3: $NoFTG_{SHAM}$, $FTG_{SHAM}$, $FTG_{M1}$; two levels for Experiment 4: $NoFTG_{SEQUENCE}$, $FTG_{SEQUENCE}$). For testing long-term effects of fatigue without training, MVC before and after the task ($\Delta$MVC(pre-post)) was compared to determine the induction of fatigue using a paired two-tailed Student's t-test.

## Induction of Fatigue

### Experiment 1

To evaluate if fatigue was reliably induced over the whole course of the task in the fatigue condition, we performed an rmANOVA with the within-subject factor time (five levels: preB1, preB2, preB3, preB4, postB4). On day 1, the fatigued group showed a significant drop of MVC over time $F(4,76) = 6.46$, $p<0.001$. In the NoFTG group paired comparisons between MVC before and after the task showed no significant difference ($p=0.16$). On day 2, no difference in MVC for the two groups was found (time ($F(1,34) = 2.90$, $p=0.10$; group ($F(1,34) = 1.75$, $p=0.20$; time*group ($F(1,34) = 0.19$, $p=0.67$).

### Experiment 2

For experiment 2 we found a significant interaction of time*group for the right hand ($F(1,18) = 4.823$, $p=0.041$). There was a significant difference for $\Delta$MVC_right between the $NoFTG_{TRANSFER}$ and $FTG_{TRANSFER}$ ($t(1,18) = -2.2$, $p=0.041$, $NoFTG_{TRANSFER}$ $-10$ Newton $\pm$ 7.18 indicating higher MVC for the control group after the task, $FTG_{TRANSFER}$ 7.77 Newton $\pm$ 3.73 indicating lower forces post task for the fatigue group as expected). No significant interaction of time*group nor an significant difference of $\Delta$ MVC_left between groups was found ($F(1,18) = 0.789$, $p=0.748$; $t(1,18) = 0.328$, $p=0.747$).

### Experiment 3

Looking at ΔMVC(pre-post), we found an significant interaction between day*group (F(2,37) = 3.390, p=0.044). On Day one the ΔMVC of the control group was different from both fatigued groups, while no significant difference was found between them (F(2,37) = 9.755, p<0.001, NoFTG vs FTG$_{SHAM}$, t(2,37) = −3.49, p=0.004, vs FTG$_{M1}$, t(2,37) = −4.27, p<0.001, FTG$_{SHAM}$ vs FTG$_{M1}$, t(2,37) = −0.87, p=0.9, NoFTG$_{SHAM}$−7.32 Newton ± 3.64, FTG$_{SHAM}$ 6.9 Newton ± 1.97, FTG_M1 10 Newton ± 2.81). On Day two no significant difference in ΔMVC between all groups was found (F(2,37) = 1.631, p=0.209).

### Experiment 4

Looking at ΔMVC(pre-post), we found a significant interaction between day*group (F(1,17) = 7.151, p=0.017). On Day one the ΔMVC of the control group was different from the fatigued group (t(=1,17)=0.459, p=0.006. In the NoFTG$_{SEQUENCE}$ group paired comparisons between MVC before and after the task showed no significant difference (t(1,8) = −0.491 p=0.639), while the FTG$_{SEQUENCE}$ group showed significant decline of force (t(1,8) = 3.341 p=0.01. On Day two no significant difference in ΔMVC was found (t(1,17) = 0.738, p=0.01).

## Control Condition

For the test of long-term effects of fatigue without training, subjects only completed the fatigue condition without the task on Day one and MVC significantly dropped pre to post, while no difference was found on Day two were subjects only completed the task without the fatigue condition (Day 1 t(1,4) = 2.92, p=0.043; Day 2 t(1,4) = 1.91, p=0.129), for study design *Appendix 1—figure 6*.

## Movement Times and Error Rates

The implemented skill measure consists of two components: speed and accuracy of the movement. Because speed and accuracy are linked inversely (lower speed allows for more accuracy and vice versa), comparing accuracy at different speed level can be difficult to interpret (*Hardwick et al., 2017*). We separately looked at the two components of the implemented skill measure to determine if differences between the fatigued and the control group were based on changes in speed, accuracy or both.

### Experiment 1

Results showed that differences in the skill measure between the fatigued and the control group were based on divergent error rates while movement time stayed similar. Participants got faster from B1 to B4 but there was no significant difference between groups or an interaction between block*groups on both days (Day 1: block: F(3,108) = 90.692, p<0.001; group: (F(1,36) = 0.019, p=0.892), block*group: (F(3,36) = 0.220, p=0.760; Day 2: block: F(3,108) = 18.425, p<0.001; group: (F(1,36) = 0.569, p=0.456), block*group: (F(3,36) = 1.423, p=0.240). Regarding error rate, there was significant effect of 'group' (F(1,36) = 2.072, p=0.031) and no significant effects of 'block' and 'block*group' on Day 1 (F(3,108) = 8.53, p=0.44; F(3,36) = 0.388, p=0.7). On Day 2, we found significant effect of 'block' (F(3,108) = 3.827, p=0.012) and no significant effect of 'group' or 'block*group' (F(1,36) = 2.155, p=0.151; (F(3,36) = 0.511, p=0.675). To be better understand the origin of the lower skill rate of the fatigued participants, we additionally analyzed the applied forces of each group. Surprisingly, the fatigued group exerted overall more force than the control group (t(1,36) = −2.61, p=0.013, NoFTG 6.9 Newton ± 0.012, FTG 7.8 Newton ± 0.048). Interestingly overshooting was only significantly different between groups for the lower force targets T1 and T4 (t(1,36) = −3.172, p=0.003, respectively t(1,36) = −2.537, p=0.015) but not the higher force targets T2 and T3 (t(1,36) = −1.036, p=0.307, respectively t(1,36) = −2.104, p=0.051).

## Experiment 2

Movement time was similar between both groups for the right and the left hand (Right hand: $F_{(1,18)}$ = 1.975, p=0.177; Left hand: $F_{(1,18)}$ = 0.392, p=0.539), while error rates were significantly different (Right hand: $F_{(1,18)}$ = 26.075, p<0.001; Left hand: $F_{(1,18)}$ = 29.555, p<0.001).

## Experiment 3

As expected, in experiment 3 we found differences in error rate but not movement time between groups (movement time/Day 1: $F_{(2,37)}$ = 0.817, p=0.449;/day 2: $F_{(2,37)}$ = 0.075; error rate/Day 1: $F_{(2,37)}$ = 9.358, p=0.001; Day 2: $F_{(2,37)}$ = 4.995, p=0.012). There was no interaction between block*group, $F_{(6,37)}$ = 0.175, p=0.983).

**Appendix 1—table 1.** error rate experiment 3.

| Group | Day 1 | | | | Day 2 | | | |
|---|---|---|---|---|---|---|---|---|
| | B1 | B2 | B3 | B4 | B1 | B2 | B3 | B4 |
| NoFTG_sham | 0.58 ± 0.05 | 0.38 ± 0.05 | 0.34 ± 0.06 | 0.29 ± 0.06 | 0.44 ± 0.04 | 0.35 ± 0.04 | 0.31 ± 0.05 | 0.23 ± 0.04 |
| FTG_sham | 0.78 ± 0.04 | 0.70 ± 0.04 | 0.61 ± 0.05 | 0.56 ± 0.05 | 0.59 ± 0.03 | 0.48 ± 0.04 | 0.44 ± 0.04 | 0.39 ± 0.03 |
| FTG_M1 | 0.68 ± 0.04 | 0.57 ± 0.04 | 0.49 ± 0.05 | 0.5 ± 0.05 | 0.52 ± 0.03 | 0.43 ± 0.04 | 0.38 ± 0.04 | 0.30 ± 0.03 |

DOI: https://doi.org/10.7554/eLife.40578.011

# Excitability changes related to fatigue and DePo protocol

Using TMS we analyzed changes of cortical excitability, depicted by motor evoke potential amplitudes (MEP) pre-, post-training and post_DePo protocol. The control group did not show significant changes between pre, post and post_DePo measurements ($F_{(2,18)}$=2.780, p=0.089). For both fatigued groups we found a differences between pre, post and post_DePo measurements (FTG_M1: $F_{(2,37)}$=7.536, p=0.007; FTG_sham: $F_{(2,37)}$=5.919, p=0.017). This was due to markedly reduced post-task MEP amplitudes (FTG_M1 pre vs post $t_{(14)}$= 5.321; p<0.001; FTG_sham pre vs post $t_{(14)}$= 3.795; p=0.002), see *Appendix 1—figure 1*. This is in line with previous reports of MEP amplitude depression after fatiguing exercise (*Gruet et al., 2013*). In both groups we found no significant difference between post and post_Depo MEP amplitudes, nor between pre and post_Depo amplitudes. This finding was expected as the specific depotentiation protocol of TMS has been shown to reverse LTP/LTD-like effects after learning without changing motor cortex excitability if administered alone (*Cantarero et al., 2013a*; *Huang et al., 2010*).

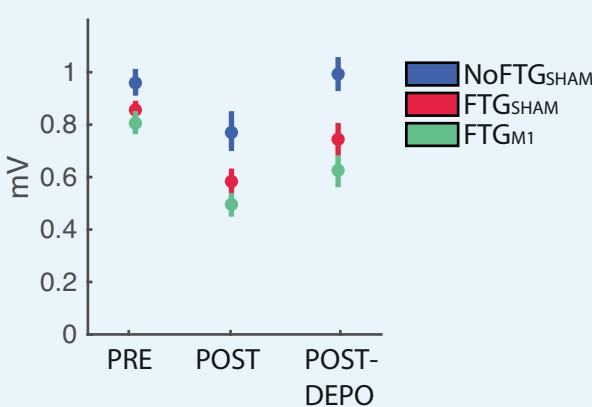

**Appendix 1—figure 1.** Changes of M1 excitability in experiment 3.
DOI: https://doi.org/10.7554/eLife.40578.012

| Group | Pre | Post | Post_DePo |
|---|---|---|---|
| NoFTG_sham | 0.96 ± 0.1 | 0.79 ± 0.1 | 0.99 ± 0.12 |
| FTG_sham | 0.86 ± 0.05 | 0.58 ± 0.08 | 0.74 ± 0.11 |
| FTG_M1 | 0.81 ± 0.08 | 0.49 ± 0.05 | 0.61 ± 0.09 |

## Control Condition

To test possible long-lasting effects of fatigue on performance rather than learning, we induced fatigue on Day one as previously described, but participants were not trained on the skill task. Instead, they took breaks matched to the time the other groups needed to finish one block on average (FTG_{SKILL-DAY2}, $N = 5$). On Day 2, this group showed similar learning rates when compared to the control group on Day 1 (mean slope FTG_{SKILL-DAY2} 0.049; p=0.50; *Appendix 1—figure 2*), see *Appendix 1—figure 2*.

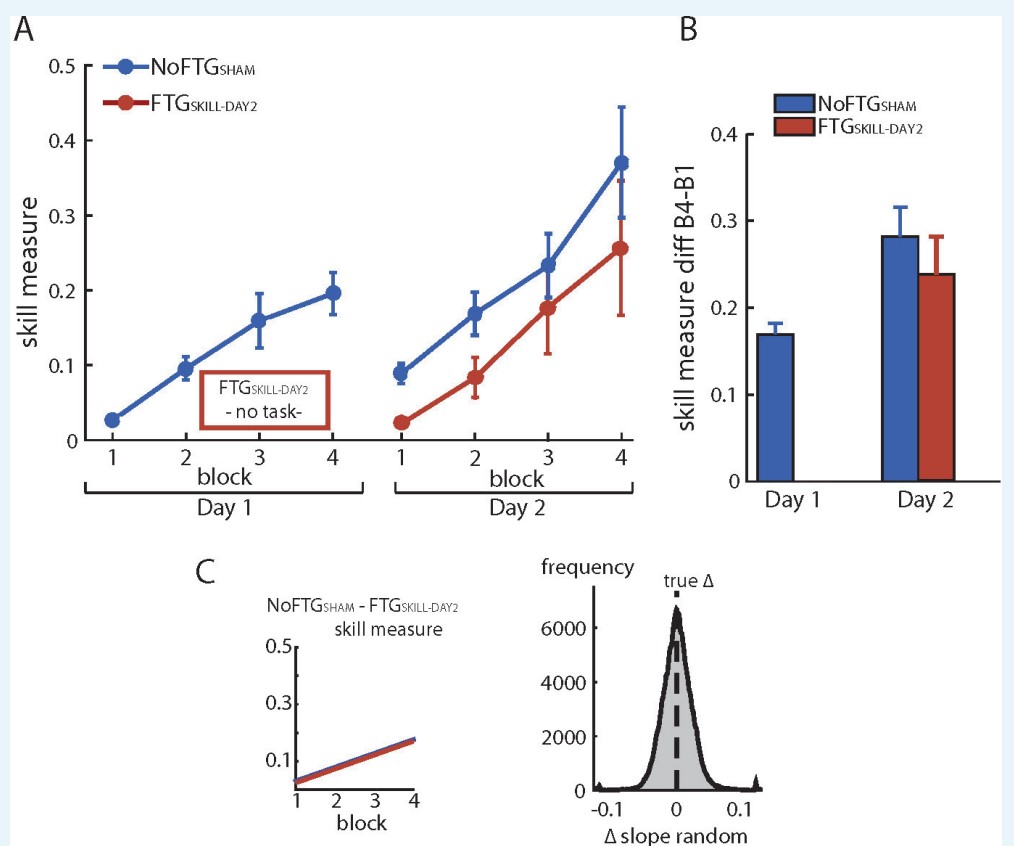

**Appendix 1—figure 2.** Fatiguing without training did not affect learning on Day 2. Panel (**A**) shows changes in skill measure over the course of four blocks on two consecutive days for the non-fatigued group (NoFTG$_{SHAM}$, same control group as in experiment 3) and on Day two for the FTG$_{SKILL-DAY2}$ (participants were fatigued on Day 1, but did not learn the task until Day two when they trained in the absence of fatigue). FTG$_{SKILL-DAY2}$ learned the task on Day two to a similar extent as the non-fatigued group on Day 1. As expected, the FTG$_{SKILL-DAY2}$ performed at a lower level compared to the non-fatigued group on Day 2. Panel (**B**) shows the difference in performance between block 4 to block one for both groups across days, the relative improvement in the FTG$_{SKILL-DAY2}$ was comparable to the control group on both days. Panel (**C**) shows the learning rates for the control group on Day one compared to the FTG$_{SKILL-DAY2}$. No differences in learning rates were found using permutation testing.
DOI: https://doi.org/10.7554/eLife.40578.014

# Study Design

## Visualization of study designs

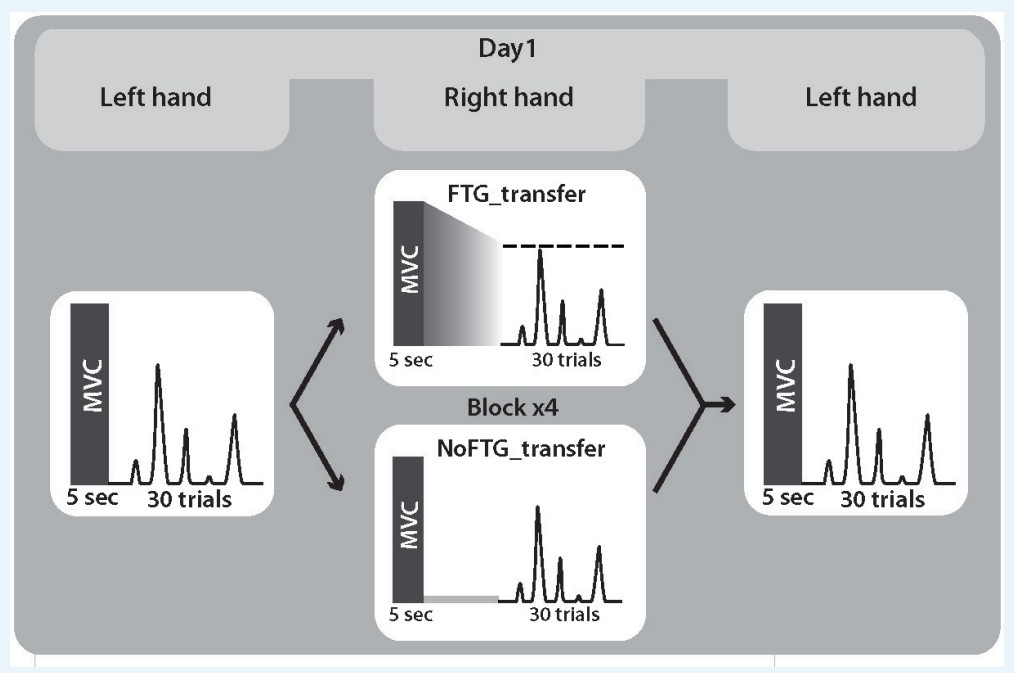

**Appendix 1—figure 3.** Study design experiment 2.

DOI: https://doi.org/10.7554/eLife.40578.015

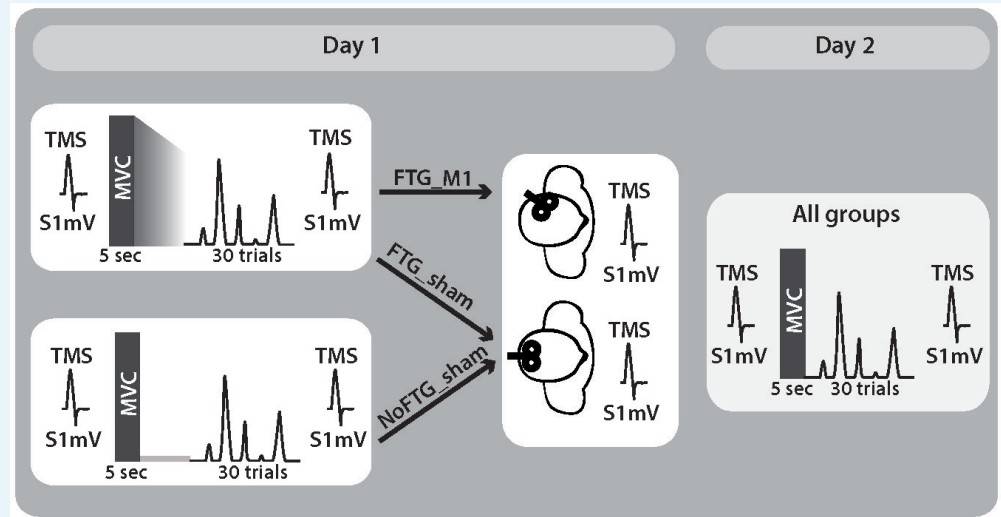

**Appendix 1—figure 4.** Study design experiment 3.

DOI: https://doi.org/10.7554/eLife.40578.016

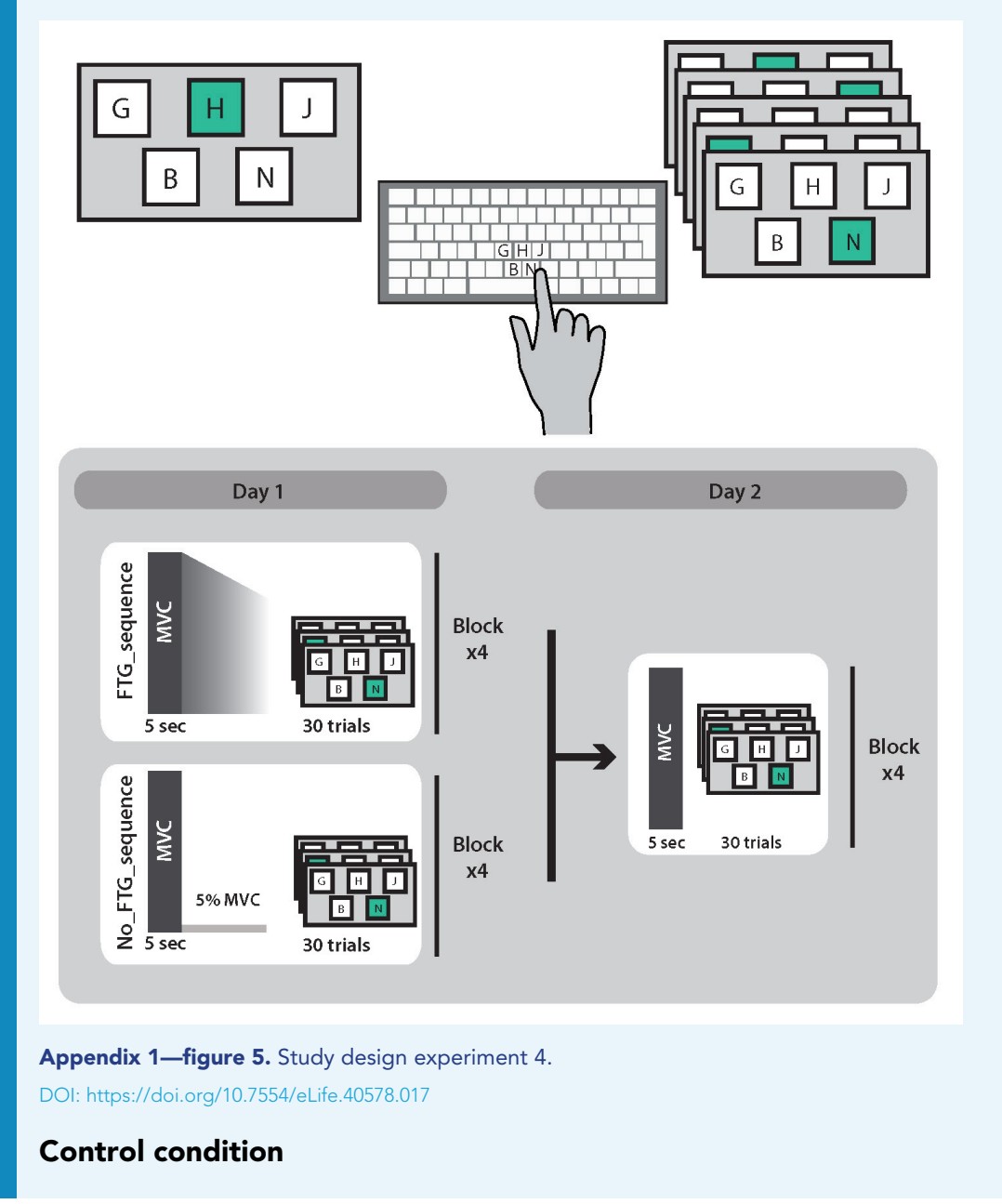

**Appendix 1—figure 5.** Study design experiment 4.

DOI: https://doi.org/10.7554/eLife.40578.017

## Control condition

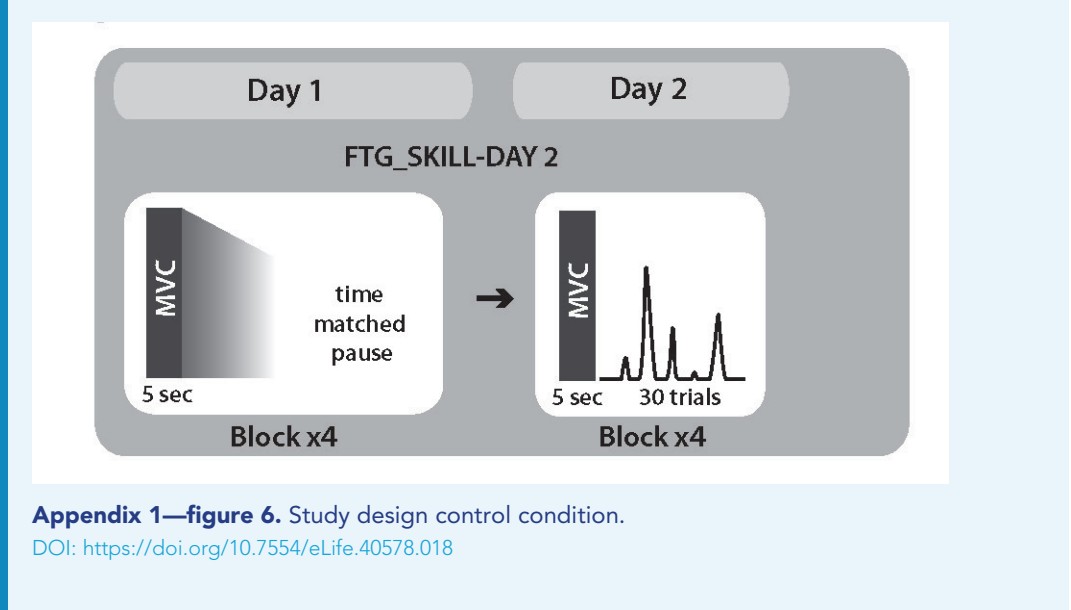

**Appendix 1—figure 6.** Study design control condition.
DOI: https://doi.org/10.7554/eLife.40578.018

