## [Decision Letter]

Thank you for submitting your article "Fatigue induces long lasting detrimental changes in motor skill learning" for consideration by *eLife*. Your article has been reviewed by Simon Gandevia (Reviewer #1) and Anna Kuppuswamy (Reviewer #2), and the evaluation has been overseen by Heidi Johansen-Berg as the Reviewing Editor and Richard Ivry as the Senior Editor.

The reviewers have discussed the reviews with one another and the Reviewing Editor has drafted this decision to help you prepare a revised submission.

Summary:

Branscheidt and colleagues have systematically investigated the effect of fatigue on motor learning in a series of 3 experiments. In the first, they show that fatigue reduces the rate at which 'skill' is acquired during a subsequent training session. They also show that the skill acquisition remains suppressed in day 2 of training in fatigue group, despite no fatigue inducing protocol. In experiment 2 they show that performance in the non-fatigued hand after training in the fatigued hand is different between the fatigue and non-fatigue group. In experiment 3 they show that disruptive rTMS at the end of day 1 improves skill acquisition in day 2 in the fatigued group. however the rate of learning does not reach the rates at which the non-fatigued groups learn. In a control experiment they show that simply inducing fatigue without subsequent motor learning does not impact on the rate of skill acquisition in day 2 of training suggesting that the combination of training under fatigue is detrimental and not just fatigue on its own.

Essential revisions:

1) The need for a control condition.

Both reviewers felt that an additional control condition is required to support the authors' conclusions that the observed effects are related to force. From the individual reviews:

Reviewer 1: Subsection “Fatigue has lasting effects on acquisition of motor skill”, third paragraph: The unexpected finding that there was increased force production in the fatigue group is quite odd. While many of the other findings in the submission are competently replicated, it would be interesting to know more about this finding. Note that across the literature on force tasks there have been some anomalous reports: usually a force generated by a weakened muscle is overestimated but there are a couple of exceptions to this. In terms of one key finding, it would be helpful to know if the following scenario has been dealt with by the many control studies. If instead of a force-based task, some other mentally challenging task had been undertaken, would the results have been the same? This is potentially a non-trivial question and I appreciate that the authors have thought about it and may consider it one of the 'to-do' tasks. However, the claims in the manuscript would be substantially altered if other sorts of task produced the same decrement. I need some convincing on this point please.

Reviewer 2: MVC drops to approx. 40% of original MVC before start of training. Subsection “Experiment 1: Determining the effect of fatigue on temporal aspects of motor skill learning” states that the force required to carry out task was intact. However, I am curious to know if the authors think that a task that requires 100% MVC will be learnt at the same rate as a task that requires less than 50% MVC. This is especially relevant as you also find that it is the difference in errors and not movement speed that affects the skill measure. We know that at higher force levels to maintain a steady force is more difficult due to higher variability i.e. errors. I suggest a control experiment where task parameters are recalibrated to new MVC, alternatively, maybe some past studies to support the idea that despite differences in task parameters, rate of learning remains the same in a given task.

After consultations, the reviewers and editor concluded that the addition of a single control condition, that requires use of the fatigued muscle and is cognitively challenging but is not force-based, would address these concerns (e.g., the addition of a sequence learning condition).

2) In experiment 2, I am slightly confused by the comparisons. Am I right in thinking that you compare across groups in time points to make your point of transfer being different between fatigued and non-fatigued groups. If this is true, then the conclusion must be that the transfer of skill to the non-fatigued hand is no different between fatigued and non-fatigued group. The transfer itself would be considered different only if the two groups behaved differently in terms of how much of the skill transferred to the non-fatigued hand. Please clarify what tests were performed and how 'transfer' has been defined and results interpreted.

3) Subsection “Motor task”: Note that a pinch task is complicated from a muscle mechanical point of view. The much stronger thumb flexor compartment is opposed to a less strong index flexor task group. Simplistically, this is likely to mean that the fatigue effected the "weaker" muscle group. While this sort of task is useful for looking at behavioural measures, it is not helpful for looking at mechanisms related to peripheral and central aspects of fatigue. A small acknowledgement of this complication would be good.

4) In the TMS experiment, do we know that all those in the real group did actually alter cortical excitability after TMS? We know that some people don't respond or respond differently to rTMS protocols. It would be useful to add the change in M1 excitability post rTMS protocol on day 1 in the Materials and methods.

---

## [Author Response]

Essential revisions:1) The need for a control condition.Both reviewers felt that an additional control condition is required to support the authors' conclusions that the observed effects are related to force. From the individual reviews:Reviewer 1: Subsection “Fatigue has lasting effects on acquisition of motor skill”, third paragraph: The unexpected finding that there was increased force production in the fatigue group is quite odd. While many of the other findings in the submission are competently replicated, it would be interesting to know more about this finding. Note that across the literature on force tasks there have been some anomalous reports: usually a force generated by a weakened muscle is overestimated but there are a couple of exceptions to this. In terms of one key finding, it would be helpful to know if the following scenario has been dealt with by the many control studies. If instead of a force-based task, some other mentally challenging task had been undertaken, would the results have been the same? This is potentially a non-trivial question and I appreciate that the authors have thought about it and may consider it one of the 'to-do' tasks. However, the claims in the manuscript would be substantially altered if other sorts of task produced the same decrement. I need some convincing on this point please.

The finding that force production in the fatigued group was increased came to our surprise too. As with the other results, we replicated this observation in all experiments (1, 2 and 3). Of note, higher force production was only detected for the lower force targets. We therefore interpreted these results as an issue of force control/perception. In support of our findings, Park et al. found in a finger force-matching task, that after fatigue force production was higher compared to the instructed reference force at low force levels (15% of MVC) but not for medium force levels (35% MVC), though this was not significant across all conditions (different finger combinations) (Park, Leonard, and Li, 2007). We added this supporting evidence to the second paragraph of the Discussion.

The second concern raised by reviewer 1 is addressed in our comments related to the new control condition added to this submission.

Reviewer 2: MVC drops to approx. 40% of original MVC before start of training. Subsection “Experiment 1: Determining the effect of fatigue on temporal aspects of motor skill learning” states that the force required to carry out task was intact. However, I am curious to know if the authors think that a task that requires 100% MVC will be learnt at the same rate as a task that requires less than 50% MVC. This is especially relevant as you also find that it is the difference in errors and not movement speed that affects the skill measure. We know that at higher force levels to maintain a steady force is more difficult due to higher variability i.e. errors. I suggest a control experiment where task parameters are recalibrated to new MVC, alternatively, maybe some past studies to support the idea that despite differences in task parameters, rate of learning remains the same in a given task.After consultations, the reviewers and editor concluded that the addition of a single control condition, that requires use of the fatigued muscle and is cognitively challenging but is not force-based, would address these concerns (e.g., the addition of a sequence learning condition).

We agree with the reviewer that performance of a task at maximal force compared to f. e. 50% of MVC might lead to differences in task difficulty and consequently might not result in the same learning rates. However, we believe that the task performance on Day 2 in the absence of fatigue complies with the control condition that reviewer 2 suggested. On Day 2, both groups show no significant differences in their MVCs; indeed, they are performing with similar total force capacity (see supplemental material). Despite this, and the fact that they have been familiarized with the task already, participants from the fatigued group still show lower learning rates than controls on Day 1. We have now emphasized the fact that on Day 2 controls and the FTG group performed the task within the same force range in the last paragraph of the subsection “Muscle fatigue has lasting effects on acquisition of force-control demanding motor skill”.

As suggested by the reviewers and the editor, we conducted a new control experiment testing whether motor fatigue also interferes with learning a more cognitively demanding task. Eighteen participants were recruited to learn a cognitive demanding 10-sequence element task, that had low force-control demand. The group was divided and randomly assigned to fatigue or non-fatigue as done in experiment 1. Participants were presented with a horizontal display of five square stimuli (‘G’, ‘H’,’J’, ‘B’ or ‘N’) with one highlighted in green and had to press the corresponding computer key on a desktop keyboard with their index finger. The next element of the sequence was only presented after the correct key response. If an incorrect key response was pressed, the highlighted green box did not advance to the next element until the appropriate key press was made. Each sequence trial started with the presentation of a “go” cue. Participants were exposed to the same 10-element sequence on each trial and had to perform 30 trials in total to complete one of four blocks. As in the other experiments, we recorded movement time (duration between first and final element response) and error rate (number of errorless sequences per block). (Appendix 1—figure 5).

We found that both groups had a similar performance of the sequence learning task on both days. There were no significant differences in movement time or error rate (Figure 5).

We added a description of the experiment and the results to the third paragraph of the Introduction and subsection “Fatigue in the absence of training does not impair learning on a subsequent day”. Additional descriptions of the study design and methods were added in the Materials and methods and Appendix 1.

In our view, this control helps qualify our conclusions indicating that the detrimental effects of muscle fatigue on learning are specific to skill tasks that required fine force-control, but not in more cognitive-demanding tasks. Thus, we argue that muscle fatigue leads to changes in the pattern and intensity of muscle activation leading to forming specific memories that are not ideal for continued learning when they are recalled under non-fatigue conditions, for instance during subsequent training.

To underline that this effect is probably limited to force-based task we changed the wording of the second paragraph of the Discussion and the conclusion at subsection “Design” – “Experiment 1: Determining the effect of fatigue on temporal aspects of motor skill learning”.

2) In experiment 2, I am slightly confused by the comparisons. Am I right in thinking that you compare across groups in time points to make your point of transfer being different between fatigued and non-fatigued groups. If this is true, then the conclusion must be that the transfer of skill to the non-fatigued hand is no different between fatigued and non-fatigued group. The transfer itself would be considered different only if the two groups behaved differently in terms of how much of the skill transferred to the non-fatigued hand. Please clarify what tests were performed and how 'transfer' has been defined and results interpreted.

We appreciate the reviewer’s confusion. As the reviewer points out, we found no clear evidence of transfer abnormalities across groups. Thus, we leverage the inter-manual skill transfer as an assay for learning rates on Day 1 to circumvent the performance confounder. While differences in skill between the control group and the fatigued group could arise from impeded execution in the fatigued hand this could mask the fact that fatigued participants were actually learning just as much about the task as their non-fatigued counterparts. If fatigue simply led to impaired execution but yielded similar learning rates, then performance of the non-fatigued effector in both groups should have been similar. We clarify this point in the subsection “Long lasting detrimental effects of fatigue on learning are centrally mediated”.

3) Subsection “Motor task”: Note that a pinch task is complicated from a muscle mechanical point of view. The much stronger thumb flexor compartment is opposed to a less strong index flexor task group. Simplistically, this is likely to mean that the fatigue effected the "weaker" muscle group. While this sort of task is useful for looking at behavioural measures, it is not helpful for looking at mechanisms related to peripheral and central aspects of fatigue. A small acknowledgement of this complication would be good.

We agree with the reviewers and underlined the limitation of the chosen behavioural task regarding the involvement of different muscle groups and to discern peripheral and central effects of fatigue (see subsection “Motor task”).

4) In the TMS experiment, do we know that all those in the real group did actually alter cortical excitability after TMS? We know that some people don't respond or respond differently to rTMS protocols. It would be useful to add the change in M1 excitability post rTMS protocol on day 1 in the Materials and methods.

We agree with the reviewer that prior studies have shown that not all subjects respond the same way to rTMS protocols. However, we would like to point out that changes of cortical excitability are not necessarily expected after the depotentiation protocol used here. This specific protocol of TMS has been shown to reverse LTP/LTD-like effects after learning (Cantarero, Lloyd, and Celnik, 2013). Changes of motor cortex excitability are not observed if depotentiation stimulation is administered alone or after motor cortex excitability is already depressed (Huang et al., 2010).

Still, as requested by the reviewers, we analyzed changes of cortical excitability after TMS. The control group did not show significant changes between pre, post and post_DePo measurements (F(2,18)=2.780, p=0.089). For both fatigued groups we found a differences between pre, post and post_DePo measurements (FTG_M1: F(2,37)=7.536, p=0.007; FTG_sham: F(2,37)=5.919, p=0.017). This was due to markedly reduced post task MEP amplitudes (FTG_M1 pre vs post t(14)= 5.321; p<0.001; FTG_sham pre vs post t(14)= 3.795; p=0.002). This is in line with previous reports of MEP amplitude depression after fatiguing exercise (Gruet et al., 2013). In both groups we found no significant difference between post and post_Depo MEP amplitudes, nor between pre and post_Depo amplitudes. We added this analysis to Appendix 1—figure 1.